# A Quantum Heat Exchanger for Nanotechnology

**DOI:** 10.3390/e22040379

**Published:** 2020-03-26

**Authors:** Amjad Aljaloud, Sally A. Peyman, Almut Beige

**Affiliations:** 1The School of Physics and Astronomy, University of Leeds, Leeds LS2 9JT, UK; S.Peyman@leeds.ac.uk (S.A.P.); a.beige@leeds.ac.uk (A.B.); 2Department of Physics, College of Sciences, University of Hail, Hail PO Box 2440, Saudi Arabia; 3Leeds Institute for Medical Research, School of Medicine, Wellcome Trust Brenner Building, St James’ Teaching Hospital, University of Leeds, Leeds LS9 7TF, UK

**Keywords:** quantum thermodynamics, laser cooling, cavitation, sonoluminescence

## Abstract

In this paper, we design a quantum heat exchanger which converts heat into light on relatively short quantum optical time scales. Our scheme takes advantage of heat transfer as well as collective cavity-mediated laser cooling of an atomic gas inside a cavitating bubble. Laser cooling routinely transfers individually trapped ions to nano-Kelvin temperatures for applications in quantum technology. The quantum heat exchanger which we propose here might be able to provide cooling rates of the order of Kelvin temperatures per millisecond and is expected to find applications in micro- and nanotechnology.

## 1. Introduction

Since its discovery in 1975 [1,2], laser cooling of individually trapped atomic particles has become a standard technique in quantum optics laboratories worldwide [3,4]. Rapidly oscillating electric fields can be used to strongly confine charged particles, such as single ions, for relatively large amounts of time [5]. Moreover, laser trapping provides unique means to control the dynamics of neutral particles, such as neutral atoms [6,7]. To cool single atomic particles, laser fields are applied which remove vibrational energy at high enough rates to transfer them down to near absolute-zero temperatures [5]. Nowadays, ion traps are used to perform a wide range of high-precision quantum optics experiments. For example, individually trapped ions are at the heart of devices with applications in quantum technology, such as atomic and optical clocks [8,9], quantum computers [10,11,12,13], quantum simulators [14,15] and electric and magnetic field sensors [16].

For laser cooling to be at its most efficient, the confinement of individually trapped particles should be so strong that the quantum characteristics of their motion is no longer negligible. This means that their vibrational energy is made up of energy quanta, which have been named phonons. When this applies, an externally applied laser field not only affects the electronic states of a trapped ion, but it also changes its vibrational state. Ideally, laser frequencies should be chosen such that the excitation of the ion should be most likely accompanied by the loss of a phonon. If the ion returns subsequently into its ground state via the spontaneous emission of a photon, its phonon state remains the same. Overall one phonon is permanently lost from the system which implies cooling. On average, every emitted photon lowers the vibrational energy of the trapped ion by the energy of one phonon. Eventually, the cooling process stops when the ion no longer possesses any vibrational energy.

Currently, there are many different ways of designing and fabricating ion traps [17,18]. However, the main requirements for the efficient conversion of vibrational energy into light on relatively short quantum optical time scales are always the same [19,20]:(1)Individual atomic particles need to be so strongly confined that the quantum character of their motion has to be taken into account. In the following, ν denotes the phonon frequency and ℏν is the energy of a single phonon.(2)A laser field with a frequency $\omega_{L}$ below the atomic transition frequency $\omega_{0}$ needs to be applied. As long as the laser detuning $\Delta =\omega_{0}-\omega_{L}$ and the phonon frequency ν are comparable in size,
(1)$$\begin{array}{ccc} \Delta & ∼ & \nu \;, \end{array}$$
the excitation of an ion is more likely accompanied by the annihilation of a phonon than by the creation of a phonon. Transitions which result in the simultaneous excitation of an ion and the creation of a phonon are possible but are less likely to occur as long as their detuning is larger.(3)When excited, the confined atomic particle needs to be able to emit a photon. In the following, we denote its spontaneous decay rate by Γ. This rate should not be much larger than ν,
(2)$$\begin{array}{ccc} \nu & \geq & \Gamma \;, \end{array}$$
so that the cooling laser couples efficiently to atomic transitions. At the same time, Γ should not be too small so that de-excitation of the excited atomic state happens often via the spontaneous emission of a photon.

Given these three conditions, the applied laser field results in the conversion of the vibrational energy of individually trapped ions into photons. As mentioned already above, laser cooling can prepare individually trapped atomic particles at low enough temperatures for applications in high-precision quantum optics experiments and in quantum technology.

In this paper, we ask the question whether laser cooling could also have applications in micro-and nanoscale physics experiments. For example, nanotechnology deals with objects which have dimensions between 1 and 1000 nm and is well known for its applications in information and communication technology, as well as sensing and imaging. Increasing the speed at which information can be processed and the sensitivity of sensors is usually achieved by reducing system dimensions. However, smaller devices are usually more prone to heating as thermal resistances increase [21]. Sometimes, large surface to volume ratios can help to off-set this problem. Another problem for nanoscale sensors is thermal noise. As sensors are reduced in size, their signal to noise ratio usually decreases and thus the thermal energy of the system can limit device sensitivity [22]. Therefore, thermal considerations have to be taken into account and large vacuums or compact heat exchangers have already become an integral part of nanotechnology devices.

Usually, heat exchangers in micro- and nanotechnology rely on fluid flow [23]. In this paper, we propose an alternative approach. More concretely, we propose to use heat transfer as well as a variation of laser cooling, namely cavity-mediated collective laser cooling [24,25,26,27,28], to lower the temperature of a small device. As illustrated in Figure 1, the proposed quantum heat exchanger mainly consist of a liquid which contains a large number of cavitating bubbles filled with noble gas atoms. Transducers constantly change the radius of these bubbles which should resemble optical cavities when they reach their minimum radius during bubble collapse phases. At this point, a continuously applied external laser field rapidly transfers vibrational energy of the atoms into light. If the surrounding liquid contains many cavitating bubbles, their surface area becomes relatively large and there can be a very efficient exchange of heat between the inside and the outside of cavitating bubbles. Any removal of thermal energy from the trapped atomic gas inside bubbles should eventually result in the cooling of the surrounding liquid and of the surface area of the device on which it is placed.

In this paper, we emphasize that cavitating bubbles can provide all of the above listed requirements for laser cooling, especially a very strong confinement of atomic particles, such as nitrogen [29,30]. For example, calculations based on a variation of the Rayleigh–Plesset equations show that the pressure at the location of a cavitating bubble can be significantly larger than the externally applied driving pressure [31]. However, the strongest indication for the presence of phonon modes with sufficiently large frequencies for laser cooling to work comes from the fact that sonoluminescence experiments are well-known for converting sound into relatively large amounts of thermal energy, while producing light in the optical regime [32,33,34]. During this process, the atomic gas inside a cavitating bubbles can reach very high temperatures [35,36], which hints at very strong couplings between electronic and vibrational degrees of freedom. In addition, the surfaces of cavitating bubbles can become opaque during the bubble collapse phase [37], thereby creating a spherical optical cavity [38,39] which is an essential requirement for cavity-mediated collective laser cooling.

To initiate the cooling process, an appropriately detuned laser field needs to be applied in addition to the transducers which confine the bubbles with sound waves. Although sonoluminescence has been studied in great detail and the idea of applying laser fields to cavitating bubbles is not new [40], not enough is known about the relevant quantum properties, such as phonon frequencies. Hence, we cannot predict realistic cooling rates for the experimental setup shown in Figure 1. A crude estimate which borrows data from different, already available experiments suggests that it might be possible to achieve cooling rates of the order of Kelvin temperatures per millisecond for volumes of liquid on a cubic micrometer scale. Cavitating bubbles already have applications in sonochemistry, where they are used to provide energy for chemical reactions [41]. Here, we propose to exploit the atom–phonon interactions in sonoluminescence experiments for laser cooling. In the presence of an appropriately detuned laser field, we expect other, highly-detuned heating processes to become secondary.

There are five sections in this paper. The purpose of Section 2 is to provide an introduction to cavity-mediated collective laser cooling of an atomic gas. As we show below, this technique is a variation of standard laser cooling techniques for individually trapped atomic particles. We provide an overview of the experimental requirements and estimate achievable cooling rates. Section 3 studies the effect of thermalization for a large collection of atoms with elastic collisions. Section 4 reviews the main design principles of a quantum heat exchanger for nanotechnology. Finally, we summarize our findings in Section 5.

## 2. Cavity-Mediated Collective Laser Cooling

In this section, we first have a closer look at a standard laser cooling technique for an individually trapped atomic particle [19,20]. Afterwards, we review cavity-mediated laser cooling of a single atom [42,43,44,45,46] and of an atomic gas [26,27,28].

### 2.1. Laser Cooling of Individually Trapped Particles

Figure 2a shows a single two-level atom (or ion) with external laser driving inside an approximately harmonic trapping potential. Most importantly, the atom should be so strongly confined that its phonon states are no longer negligible. In the following, ν denotes the frequency of the energy quanta in the vibrational energy of the atomic particle and |m〉 is a vibrational state with exactly *m* phonons. Moreover, |g〉 and |e〉 denote the ground and the excited electronic state of the trapped particle with energy separation $\hbar \omega_{0}$. Figure 2b shows the energy level of the combined atom–phonon system with the energy eigenstates |x,m〉.

To lower the temperature of the atom, the frequency $\omega_{L}$ of the cooling laser needs to be below its transition frequency $\omega_{0}$. Ideally, the laser detuning $\Delta =\omega_{0}-\omega_{L}$ equals the phonon frequency ν (cf. Equation (Equation 1)). In addition, the spontaneous decay rate Γ of the excited atomic state should not exceed ν (cf. Equation (Equation 2)). When both conditions apply, the cooling laser couples most strongly, i.e., resonantly and efficiently, to transitions for which the excitation of the atom is accompanied by the simultaneously annihilation of a phonon. All other transitions are strongly detuned. Moreover, the spontaneous emission of a photon only affects the electronic but not the vibrational states of the atom. Hence, the spontaneous emission of a photon usually indicates the loss of one phonon. Suppose the atom was initially prepared in a state |g,m〉. Then, its final state equals |g,m−1〉. One phonon has been permanently removed from the system which implies cooling. As illustrated in Figure 2b, the trapped particle eventually reaches its ground state |g,0〉 where it no longer experiences the cooling due to off-resonant driving [19,20].

To a very good approximation, the Hamiltonian of the atom–phonon system equals [20]
(3)$$\begin{array}{ccc} H_{I} & = & \hbar g\left(\sigma^{-}b^{†}+\sigma^{+}b\right) \end{array}$$
in the interaction picture with respect to its free energy. Here, *g* denotes the (real) atom–phonon coupling constant, while $\sigma^{+}=\left|e\rangle \langle g\right|$ and $\sigma^{-}=\left|g\rangle \langle e\right|$ are atomic rising and lowering operators. Moreover, *b* and $b^{†}$ are phonon annihilation and creation operators with $[b,b^{†}]=1$. To take into account the spontaneous emission of photons from the excited state of the atom with decay rate Γ, we describe the atom–phonon system in the following by its density matrix $\rho_{I}\left(t\right)$ with
(4)$$\begin{array}{ccc} {\dot{\rho }}_{I} & = & -\frac{i}{\hbar }\left[H_{I},\rho_{I}\right]+\Gamma \left(\sigma^{-}\;\rho_{I}\;\sigma^{+}-\frac{1}{2}\sigma^{+}\sigma^{-}\;\rho_{I}-\frac{1}{2}\rho_{I}\;\sigma^{+}\sigma^{-}\right)\;. \end{array}$$

This equation can be used to analyze the dynamics of the expectation value $\langle A_{I}\rangle =\mathrm{Tr}\left(A_{I}\rho_{I}\right)$ of observables $A_{I}$, since it implies
(5)$$\begin{array}{ccc} \langle {\dot{A}}_{I}\rangle & = & -\frac{i}{\hbar }\left[A_{I},H_{I}\right]+\Gamma \langle \sigma^{+}\;A_{I}\;\sigma^{-}-\frac{1}{2}A_{I}\;\sigma^{+}\sigma^{-}-\frac{1}{2}\sigma^{+}\sigma^{-}\;A_{I}\rangle \;. \end{array}$$

Here, we are especially interested in the dynamics of the mean phonon number $m=\langle b^{†}b\rangle$. To obtain a closed set of rate equations, we also need to study the dynamics of the population of the excited atomic state $s=\langle \sigma^{+}\sigma^{-}\rangle$ and the dynamics of the atom–phonon coherence $k_{1}=i\;\langle \sigma^{-}b^{†}-\sigma^{+}b\rangle$. Using Equation (Equation 5), one can show that
(6)$$\begin{array}{ccc} \dot{m} & = & -g\;k_{1}\;,\\ \dot{s} & = & g\;k_{1}-\Gamma \;s\;,\\ {\dot{k}}_{1} & = & 2g\left(m-s\right)-4g\;ms-\frac{1}{2}\Gamma \;k_{1} \end{array}$$
when assuming that $\langle \sigma^{+}\sigma^{-}b^{†}b\rangle =\langle \sigma^{+}\sigma^{-}\rangle \langle b^{†}b\rangle =ms$ to a very good approximation. Having a closer look at the above equations, we see that the system rapidly reaches its stationary state with $m=s=k_{1}=0$. Eventually, the atom reaches a very low temperature. More detailed calculations reveal that the final phonon *m* of the trapped atom depends on its system parameters but remains small as long as the ratio Γ/ν is sufficiently small [20]. The above cooling equations (Equation (Equation 6)) also show that the corresponding cooling rate equals
(7)$$\begin{array}{ccc} \gamma_{1\;\mathrm{atom}}^{\mathrm{standard}} & = & g^{2}/\Gamma \end{array}$$
to a very good approximation and that the cooling process takes place not on mechanical but on relatively short quantum optical time scales.

### 2.2. Cavity-Mediated Laser Cooling of a Single Atom

Suppose we want to cool a single atom whose transition frequency $\omega_{0}$ is well above the optical regime, i.e., much larger than typical laser frequencies $\omega_{L}$. In this case, it is impossible to realize the condition Δ∼ν in Equation (Equation 1). Hence, it might seem impossible to lower the temperature of the atom via laser cooling. To overcome this problem, we confine the particle in the following inside an optical resonator (cf. Figure 3) and denote the cavity state with exactly *n* photons by |n〉. Using this notation, the energy eigenstates of the atom–phonon–photon systems can be written as |x,m,n〉. Moreover, ν is again the phonon frequency, κ denotes the spontaneous cavity decay rate and $\omega_{L}$ and $\omega_{\mathrm{cav}}$ denote the laser and the cavity frequency, respectively.

In the experimental setup in Figure 3, all transitions which result in the excitation of the atom are naturally strongly detuned and can be neglected. However, the same does not have to apply to indirect couplings which result in the direct conversion of phonons into cavity photons [27,44]. Suppose the cavity detuning $\Delta_{\mathrm{cav}}=\omega_{\mathrm{cav}}-\omega_{L}$ and the phonon frequency ν are approximately the same and the cavity decay rate κ does not exceed ν,
(8)$$\begin{array}{ccc} \Delta_{\mathrm{cav}}\;∼\;\nu \;\; & \mathrm{and} & \;\;\nu \geq \;\kappa \;, \end{array}$$
in analogy to Equations (Equation 1) and (Equation 2). Then, two-step transitions which excite the atom while annihilating a phonon immediately followed by the de-excitation of the atom while creating a cavity photon become resonant and dominate the dynamics of the atom–phonon–photon system. The overall effect of these two-step transitions is the direct conversion of a phonon into a cavity photon, while the atom remains essentially in its ground state (cf. Figure 3b). When a cavity photon subsequently leaks into the environment, the phonon is permanently lost.

To model the above described dynamics, we describe the experimental setup in Figure 3 in the following by the interaction Hamiltonian [44,45]
(9)$$\begin{array}{ccc} H_{I} & = & \hbar g_{\mathrm{eff}}\left(bc^{†}+b^{†}c\right), \end{array}$$
where $g_{\mathrm{eff}}$ denotes the effective atom–cavity coupling constant and where *c* with $[c,c^{†}]=1$ is the cavity photon annihilation operator. Since the atom remains essentially in its ground state, its spontaneous photon emission remains negligible. To model the possible leakage photons through the cavity mirrors, we employ again a master equation. Doing so, the time derivative of the density matrix $\rho_{I}\left(t\right)$ of the phonon–photon system equals
(10)$$\begin{array}{ccc} {\dot{\rho }}_{I} & = & -\frac{i}{\hbar }\left[H_{I},\rho_{I}\right]+\kappa \left(c\rho_{I}c^{†}-\frac{1}{2}c^{†}c\rho_{I}-\frac{1}{2}\rho_{I}c^{†}c\right) \end{array}$$
in the interaction picture. Hence, expectation values $\langle A_{I}\rangle =\mathrm{Tr}\left(A_{I}\rho_{I}\right)$ of phonon–photon observables $A_{I}$ evolve such that
(11)$$\begin{array}{ccc} \langle {\dot{A}}_{I}\rangle & = & -\frac{i}{\hbar }\left[A_{I},H_{I}\right]+\kappa \langle c^{†}\;A_{I}\;c-\frac{1}{2}A_{I}\;c^{†}c-\frac{1}{2}c^{†}c\;A_{I}\rangle \;, \end{array}$$
in analogy to Equation (Equation 5). In the following, we use this equation to study the dynamics of the phonon number $m=\langle b^{†}b\rangle$, the photon number $n=\langle c^{†}c\rangle$, and the phonon–photon coherence $k_{1}=i\langle bc^{†}-b^{†}c\rangle$. Proceeding as described in the previous subsection, we now obtain the rate equations
(12)$$\begin{array}{ccc} \dot{m} & = & g_{\mathrm{eff}}k_{1}\;,\\ \dot{n} & = & -g_{\mathrm{eff}}k_{1}-\kappa n\;,\\ {\dot{k}}_{1} & = & 2g_{\mathrm{eff}}\left(n-m\right)-\frac{1}{2}\kappa k_{1}\;. \end{array}$$

These describe the continuous conversion of phonons into cavity photons which subsequently escape the system. Hence, it is not surprising to find that the stationary state of the atom–phonon–photon system corresponds to $m=n=k_{1}=0$. Independent of its initial state, the atom again reaches a very low temperature. In analogy to Equation (Equation 7), the effective cooling rate for cavity-mediated laser cooling is now given by [44,45]
(13)$$\begin{array}{ccc} \gamma_{1\;\mathrm{atom}} & = & g_{\mathrm{eff}}^{2}/\kappa \;. \end{array}$$

Due to the resonant coupling being indirect, $g_{\mathrm{eff}}$ is in general a few orders of magnitude smaller than *g* in Equation (Equation 7), if the spontaneous decay rates κ and Γ are of similar size. Cooling a single atom inside an optical resonator might therefore take significantly longer. However, as we show below, this reduction in cooling rate can be compensated for by the collective enhancement of the atom–cavity interaction constant $g_{\mathrm{eff}}$ [26].

### 2.3. Cavity-Mediated Collective Laser Cooling of an Atomic Gas

Finally, we have a closer look at cavity-mediated collective laser cooling of an atomic gas inside an optical resonator [26,27]. To do so, we replace the single atom in the experimental setup in Figure 3 by a collection of *N* atoms. In analogy to Equation (Equation 9), the interaction Hamiltonian $H_{I}$ between phonons and cavity photons now equals
(14)$$\begin{array}{ccc} H_{I} & = & \sum_{i=1}^{N}\hbar g_{\mathrm{eff}}^{\left(i\right)}\left(b_{i}c^{†}+b_{i}^{†}c\right)\;, \end{array}$$
where $g_{\mathrm{eff}}^{\left(i\right)}$ denotes the effective atom–cavity coupling constant of atom *i*. This coupling constant is essentially the same as $g_{\mathrm{eff}}$ in Equation (Equation 13) and depends in general on the position of atom *i*. Moreover, $b_{i}$ denotes the phonon annihilation operator of atom *i* with $\left[b_{i},b_{j}^{†}\right]=\delta_{ij}$. To simplify the above Hamiltonian, we introduce a collective phonon annihilation operator *B*,
(15)$$\begin{array}{ccc} B=\frac{\sum_{i=1}^{N}g_{\mathrm{eff}}^{\left(i\right)}\;b_{i}}{{\tilde{g}}_{\mathrm{eff}}}\;\; & \mathrm{with} & \;\;{\tilde{g}}_{\mathrm{eff}}={\left(\sum_{i=1}^{N}{\left|g_{\mathrm{eff}}^{\left(i\right)}\right|}^{2}\right)}^{1/2}, \end{array}$$
with $[B,B^{†}]=1$. Using this notation, $H_{I}$ in Equation (Equation 14) simplifies to
(16)$$\begin{array}{ccc} H_{I} & = & \hbar {\tilde{g}}_{\mathrm{eff}}\left(Bc^{†}+B^{†}c\right)\;. \end{array}$$

Notice that the effective coupling constant ${\tilde{g}}_{\mathrm{eff}}$ scales as the square root of the number of atoms *N* inside the cavity. For example, if all atomic particles couple equally to the cavity field with a coupling constant $g_{\mathrm{eff}}\equiv g_{\mathrm{eff}}^{\left(i\right)}$, then ${\tilde{g}}_{\mathrm{eff}}=\sqrt{N}\;g_{\mathrm{eff}}$. This means, in the case of many atoms, the effective phonon–photon coupling is collectively enhanced [26].

When comparing $H_{I}$ in Equation (Equation 9) with $H_{I}$ in Equation (Equation 14), we see that both Hamiltonians are essentially the same. Moreover, the density matrix $\rho_{I}$ obeys the master equation in Equation (Equation 10) in both cases. Hence, we expect the same cooling dynamics in the one atom and in the many atom case. Suppose all atoms experience the same atom–cavity coupling constant $g_{\mathrm{eff}}$, the effective cooling rate of the common vibrational mode *B* becomes
(17)$$\begin{array}{ccc} \gamma_{N\;\mathrm{atoms}} & = & Ng_{\mathrm{eff}}^{2}/\kappa \;, \end{array}$$
in analogy to Equation (Equation 13). This cooling rate is *N* times larger than the cooling rate which we predicted in the previous subsection for cavity-mediated laser cooling of a single atom. Using sufficiently large number of atoms *N*, it is therefore possible to realize cooling rates $\gamma_{N\;\mathrm{atoms}}$ with
(18)$$\begin{array}{ccc} \gamma_{N\;\mathrm{atoms}} & \gg & \gamma_{1\;\mathrm{atom}}^{\mathrm{standard}}\;. \end{array}$$

This suggests that the cooling rate of cavity-mediated laser cooling, i.e., the rate of change of the mean number *n* of *B* phonons in the system, is comparable and might even exceed the cooling rates of standard laser cooling of single trapped ions.

However, the above discussion also shows that cavity-mediated collective laser cooling only removes phonons from a single common vibrational mode *B*, while all other vibrational modes of the atomic gas do not experience the cooling laser. Once the *B* mode reaches its stationary state, the conversion of thermal energy into light stops. To nevertheless take advantage of the relatively high cooling rates of cavity-mediated collective laser cooling, an additional mechanism is needed [27,28]. As we shall see in the next section, one way of transferring energy between different vibrational modes is to intersperse cooling stages with thermalization stages (cf. Figure 4). The purpose of the cooling stages is to rapidly remove energy from the system. The purpose of subsequent thermalization stages is to transfer energy from the surroundings of the bubble and from the different vibrational modes of the atoms into the *B* mode. Repeating thermalization and cooling stages is expected to result in the cooling of the whole setup in Figure 1.

## 3. Thermalization of an Atomic Gas with Elastic Collisions

Thermalization stages occur naturally in cavitating bubbles between collapse stages due to elastic collisions. As we show below, these transfer an atomic gas into its thermal state, thereby re-distributing energy between all if its vibrational degrees of freedom. During bubble expansions, the phonon frequencies of the atoms become very small. It is therefore safe to assume that the atoms do not see the cooling laser during thermalization stages.

### 3.1. The Thermal State of a Single Harmonic Oscillator

As in the previous section, we first consider a single trapped atom inside a harmonic trapping potential. Its thermal state equals [47]
(19)$$\begin{array}{ccc} \rho =\frac{1}{Z}\;e^{-\beta H}\;\; & \mathrm{with} & \;\;Z=\mathrm{Tr}(e^{-\beta H})\;, \end{array}$$
where *H* is the relevant harmonic oscillator Hamiltonian, $\beta =1/k_{B}T$ is the thermal Lagrange parameter for a given temperature *T*, $k_{B}$ is Boltzmann’s constant and *Z* denotes the partition function which normalizes the density matrix ρ of the atom. For sufficiently large atomic transition frequencies $\omega_{0}$, the thermal state of the atom is to a very good approximation given by its ground state |g〉, unless the atom becomes very hot. In the following, we therefore neglect its electronic degrees of freedom. Hence, the Hamiltonian *H* in Equation (Equation 19) equals
(20)$$\begin{array}{ccc} H & = & \hbar \nu \left(b^{†}b+\frac{1}{2}\right)\;, \end{array}$$
where ν and *b* denote again the frequency and the annihilation operator of a single phonon. Combining Equations (Equation 19) and (Equation 20), we find that [47]
(21)$$\begin{array}{ccc} Z & = & \frac{e^{-\frac{1}{2}\lambda }}{1-e^{-\lambda }}\;. \end{array}$$
with λ=βℏν. Here, we are especially interested in the expectation value of the thermal energy of the vibrational mode of the trapped atom which equals 〈H〉=Tr(Hρ). Hence, using Equation (Equation 19), one can show that
(22)$$\begin{array}{ccc} \langle H\rangle & = & \frac{1}{Z}\mathrm{Tr}\left(He^{-\beta H}\right)=-\frac{1}{Z}\frac{\partial }{\partial \beta }Z=-\frac{\partial }{\partial \beta }\ln Z\;. \end{array}$$

Finally, combining this result with Equation (Equation 21), we find that
(23)$$\begin{array}{ccc} \langle H\rangle & = & \hbar \nu \left(\frac{e^{-\lambda }}{e^{-\lambda }-1}+\frac{1}{2}\right) \end{array}$$
which is Planck’s expression for the average energy of a single quantum harmonic oscillator. Moreover,
(24)$$\begin{array}{ccc} m & = & \frac{e^{-\lambda }}{e^{-\lambda }-1}\;, \end{array}$$
since the mean phonon number $m=\langle b^{†}b\rangle$ relates to 〈H〉 via $m=\langle H\rangle /\hbar \nu -\frac{1}{2}$.

### 3.2. The Thermal State of Many Atoms with Collisions

Next we calculate the thermal state of a strongly confined atomic gas with strong elastic collisions. This situation has many similarities with the situation considered in the previous subsection. The atoms constantly collide with their respective neighbors which further increases the confinement of the individual particles. Hence, we assume in the following that the atoms no longer experience the phonon frequency ν but an increased phonon frequency $\nu_{\mathrm{eff}}$. If all atoms experience approximately the same interaction, their Hamiltonian *H* equals
(25)$$\begin{array}{ccc} H & = & \sum_{i=1}^{N}\hbar \left(\nu_{\mathrm{eff}}+\frac{1}{2}\right)b_{i}^{†}b_{i} \end{array}$$
to a very good approximation. Here, $b_{i}$ denotes again the phonon annihilation operator of atom *i*. Comparing this Hamiltonian with the harmonic oscillator Hamiltonian in Equation (Equation 20) and substituting *H* in Equation (Equation 25) into Equation (Equation 19) to obtain the thermal state of many atoms, we find that this thermal state is simply the product of the thermal states of the individual atoms. All atoms have the same thermal state, their mean phonon number $m_{i}=\langle b_{i}^{†}b_{i}\rangle$ equals
(26)$$\begin{array}{ccc} m_{i} & = & \frac{e^{-\lambda_{\mathrm{eff}}}}{e^{-\lambda_{\mathrm{eff}}}-1} \end{array}$$
with $\lambda_{\mathrm{eff}}=\hbar \nu_{\mathrm{eff}}/k_{B}T$, in analogy to Equation (Equation 24). This equation shows that any previously depleted collective vibrational mode of the atoms becomes re-populated during thermalization stages.

## 4. A Quantum Heat Exchanger with Cavitating Bubbles

As pointed out in Section 1, the aim of this paper is to design a quantum heat exchanger for nanotechnology. The proposed experimental setup consists of a liquid on top of the device which we aim to keep cool, a transducer and a cooling laser (cf. Figure 1). The transducer generates cavitating bubbles which need to contain atomic particles and whose diameters need to change very rapidly in time. The purpose of the cooling laser is to stimulate the conversion of heat into light. The cooling of the atomic particles inside cavitating bubbles subsequently aids the cooling of the liquid which surrounds the bubbles and its environment via adiabatic heat transfers.

To gain a better understanding of the experimental setup in Figure 1, Section 4.1 describes the main characteristics of single bubble sonoluminescence experiments [32,33,34,35,36]. Section 4.2 emphasizes that there are many similarities between sonoluminescence and quantum optics experiments [29,30]. From this, we conclude that sonoluminescence experiments naturally provide the main ingredients for the implementation of cavity-mediated collective laser cooling of an atomic gas [26,27,28]. Finally, in Section 4.3 and Section 4.4, we have a closer look at the physics of the proposed quantum heat exchanger and estimate cooling rates.

### 4.1. Single Bubble Sonoluminescence Experiments

Sonoluminescence can be defined as a phenomenon of strong light emission from collapsing bubbles in a liquid, such as water [32,33,34]. These bubbles need to be filled with noble gas atoms, such as nitrogen atoms, which occur naturally in air. Alternatively, the bubbles can be filled with ions from ionic liquids, molten salts, and concentrated electrolyte solutions [48]. Moreover, the bubbles need to be acoustically confined and periodically driven by ultrasonic frequencies. As a result, the bubble radius changes periodically in time, as illustrated in Figure 5. The oscillation of the bubble radius regenerates itself with unusual precision.

At the beginning of every expansion phase, the bubble oscillates about its equilibrium radius until it returns to its fastness. During this process, the bubble temperature changes adiabatically and there is an exchange of thermal energy between the atoms inside the bubble and the surrounding liquid. During the collapse phase of a typical single-bubble sonoluminescence, i.e., when the bubble reaches its minimum radius, its inside becomes thermally isolated from the surrounding environment and the atomic gas inside the bubble becomes strongly confined. Usually, a strong light flash occurs at this point which is accompanied by a sharp increase of the temperature of the particles. Experiments have shown that increasing the concentration of atoms inside the bubble increases the intensity of the emitted light [35,36].

### 4.2. A Quantum Optics Perspective on Sonoluminescence

The above observations suggest many similarities between sonoluminescence and quantum optics experiments with trapped atomic particles [29,30]. When the bubble reaches its minimum radius, an atomic gas becomes very strongly confined [31]. The quantum character of the atomic motion can no longer be neglected and, as in ion trap experiments (cf. Section 2.1), the presence of phonons with different trapping frequencies ν has to be taken into account. Moreover, when the bubble reaches its minimum radius, its surface can become opaque and almost metallic [37]. When this happens, the bubble traps light inside and closely resembles an optical cavity which can be characterized by a frequency $\omega_{\mathrm{cav}}$ and a spontaneous decay rate κ. Since the confined particles have atomic dipole moments, they naturally couple to the quantized electromagnetic field inside the cavity. The result can be an exchange of energy between atomic dipoles and the cavity mode. The creation of photons inside the cavity is always accompanied by a change of the vibrational states of the atoms. Hence, the subsequent spontaneous emission of light in the optical regime results in a permanent change of the temperature of the atomic particles.

A main difference between sonoluminescence and cavity-mediated collective laser cooling is the absence and presence of external laser driving (cf. Section 2.3). However, even in the absence of external laser driving, there can be a non-negligible amount of population in the excited atomic states |e〉. This applies, for example, if the atomic gas inside the cavitating bubble is initially prepared in the thermal equilibrium state of a finite temperature *T*. Once surrounded by an optical cavity, as it occurs during bubble collapse phases, excited atoms can return into their ground state via the creation of a cavity photon (cf. Figure 6). Suddenly, an additional de-excitation channel has become available to them. As pointed out in Refs. [29,30], the creation a cavity photons is more likely accompanied by the creation of a phonon than the annihilation of a phonon since
(27)$$\begin{array}{ccc} B^{†} & = & \sum_{m=0}^{\infty }\sqrt{m+1}\;|m+1\rangle \langle m|\;,\\ B & = & \sum_{m=0}^{\infty }\sqrt{m}\;|m-1\rangle \langle m|\;. \end{array}$$

Here, *B* and $B^{†}$ denote the relevant phonon annihilation and creation operators, while |m〉 denotes a state with exactly *m* phonons. As one can see from Equation (Equation 27), the normalization factor of $B^{†}\;|m\rangle$ is slightly larger than the normalization factor of the state B|m〉. When the cavity photon is subsequently lost via spontaneous photon emission, the newly-created phonon remains inside the bubble. Hence, the light emission during bubble collapse phases is usually accompanied by heating, until the sonoluminescing bubble reaches an equilibrium.

During each bubble collapse phase, cavitating bubbles are thermally isolated from their surroundings. However, during the subsequent expansion phase, system parameters change adiabatically and there is a constant exchange of thermal energy between atomic gas inside the bubble and the surrounding liquid (cf. Figure 5). Eventually, the atoms reach an equilibrium between heating during bubble collapse phases and the loss of energy during subsequent expansion phases. Experiments have shown that the atomic gas in side the cavitating bubble can reaches temperature of the order of $10^{4}$ K which strongly supports the hypothesis that there is a very strong coupling between the vibrational and the electronic states of the confined particles [35,36].

### 4.3. Cavity-Mediated Collective Laser Cooling of Cavitating Bubbles

The previous subsection shows that, during each collapse phase, the dynamics of the cavitating bubbles in Figure 1 is essentially the same as the dynamics of the experimental setup in Figure 3 but with the single atom replaced by an atomic gas. When the bubble reaches its minimum diameter $d_{\min}$, it forms an optical cavity which supports a discrete set of frequencies $\omega_{\mathrm{cav}}$,
(28)$$\begin{array}{ccc} \omega_{\mathrm{cav}} & = & j\times \frac{\pi c}{d_{\min}}\;, \end{array}$$
where *c* denotes the speed of light in air and j=1,2,… is an integer. As illustrated in Figure 7, the case j=1 corresponds to a cavity photon wavelength $\lambda_{\mathrm{cav}}=2d_{\min}$. Moreover, j=2 corresponds to $\lambda_{\mathrm{cav}}=d_{\min}$, and so on. Under realistic conditions, the cavitating bubbles are not all of the same size which is why every *j* is usually associated with a range of frequencies $\omega_{\mathrm{cav}}$ (cf. Figure 7). Here, we are especially interested in the parameter *j*, where the relevant cavity frequencies lie in the optical regime. All other parameters *j* can be neglected, once a laser field with an optical frequency $\omega_{L}$ is applied, if neighboring frequency bands are sufficiently detuned.

In addition, we know that the phonon frequency ν of the collective phonon mode *B* assumes its maximum $\nu_{\max}$ during the bubble collapse phase. Suppose the cavity detuning $\Delta_{\mathrm{cav}}=\omega_{L}-\omega_{\mathrm{cav}}$ of the applied laser field is chosen such that
(29)$$\begin{array}{ccc} \Delta_{\mathrm{cav}}\;∼\;\nu_{\max}\;\; & \mathrm{and} & \;\;\nu_{\max}\;\geq \;\kappa \;, \end{array}$$
in analogy to Equation (Equation 2). As we have seen in Section 2.3, in this case, the two-step transition which results in the simultaneous annihilation of a phonon and the creation of a cavity photon becomes resonant and dominates the system dynamics. If the creation of a cavity photon is followed by a spontaneous emission, the previously annihilated phonon cannot be restored and is permanently lost. Overall, we expect this cooling process to be very efficient, since the atoms are strongly confined and cavity cooling rates are collectively enhanced (cf. Equation (Equation 17)).

To cool not only very tiny but larger volumes, the experimental setup in Figure 1 should contain a relatively large number of cavitating bubbles. Depending on the quality of the applied transducer, the minimum diameters $d_{\min}$ of these bubbles might vary in size. Consequently, the collection of bubbles supports a finite range of cavity frequencies $\omega_{\mathrm{cav}}$ (cf. Figure 7 so that it becomes impossible to realize the ideal cooling condition $\Delta_{\mathrm{cav}}∼\nu_{\max}$ in Equation (Equation 29) for all bubbles. However, as long as the frequency $\omega_{L}$ of the cooling laser is smaller than all optical cavity frequencies $\omega_{\mathrm{cav}}$, the system dynamics will be dominated by cooling and not by heating. In general, it is important that the diameters of the bubbles does not vary by too much.

Section 2.3 also shows that cavity-mediated collective laser cooling only removes thermal energy from a single collective vibrational mode *B* of the atoms. Once this mode is depleted, the cooling process stops. To efficiently cool an entire atomic gas, a mechanism is needed which rapidly re-distributes energy between different vibrational degrees of freedom, for example, via thermalization based on elastic collisions (cf. Section 3). As shown above, between cooling stages, cavitating bubbles evolve essentially adiabatically and the atoms experience strong collisions. In other words, the expansion phase of cavitating bubbles automatically implements the intermittent thermalization stages of cavity-mediated collective laser cooling.

Finally, let us point out that it does not matter whether the cooling laser is turned on or off during thermalization stages, i.e., during bubble expansion phases. As long as optical cavities only form during the bubble collapse phases, the above-described conversion of heat into light only happens when the bubble reaches its minimum diameter. The reason for this is that noble gas atoms, such as nitrogen, have very large transition frequencies $\omega_{0}$. The direct laser excitation of atomic particles is therefore relatively unlikely, even when the cooling laser is turned on. If we could excite the atoms directly by laser driving, we could cool them even more efficiently (cf. Section 2.1).

### 4.4. Cooling of the Surroundings via Heat Transfer

The purpose of the heat exchanger which we propose here is to constantly remove thermal energy from the liquid surrounding the cavitating bubbles and device on which the liquid is placed (cf. Figure 1). As described in the previous subsection, the atomic gas inside the bubbles is cooled by very rapidly converting heat into light during each collapse phase. In between collapse phases, the cavitating bubbles evolve adiabatically and naturally cool their immediate environment via heat transfer. As illustrated in Figure 8, alternating cooling and thermalization stages (or collapse and expansion phases) is expected to implement a quantum heat exchanger, which does not require the actual transport of particles from one place to another.

Finally, let us have a closer look at achievable cooling rates for micro- and nanotechnology devices with length dimensions in the nano- and micrometer regime. Unfortunately, we do not know how rapidly heat can be transferred from the nanotechnology device to the liquid and from there to the atomic gas inside the cavitating bubbles. However, any thermal energy which is taken from the atoms comes eventually from the environment which we aim to cool. Suppose the relevant phonon frequencies $\nu_{\max}$ are sufficiently large to ensure that every emitted photon indicates the loss of one phonon, i.e., the loss of one energy quantum $\hbar \nu_{\max}$. Moreover, suppose our quantum heat exchanger contains a certain amount of liquid, let us say water, of mass $m_{\mathrm{water}}$ and heat capacity $c_{\mathrm{water}}\left(T\right)$ at an initial temperature $T_{0}$. Then, we can ask the question: How many photons $N_{\mathrm{photons}}$ do we need to create in order to cool the water by a certain temperature ΔT?

From thermodynamics, we know that the change in the thermal energy of the water equals
(30)$$\begin{array}{ccc} \Delta Q & = & c_{\mathrm{water}}\left(T_{0}\right)\;m_{\mathrm{water}}\Delta T \end{array}$$
in this case. Moreover, we know that
(31)$$\begin{array}{ccc} \Delta Q & = & N_{\mathrm{photons}}\;\hbar \nu_{\max}\;. \end{array}$$

Hence, the number of photons that needs to be produced is given by
(32)$$\begin{array}{ccc} N_{\mathrm{photons}} & = & \frac{c_{\mathrm{water}}\left(T_{0}\right)\;m_{\mathrm{water}}\;\Delta T}{\hbar \nu_{\max}}\;. \end{array}$$

The time $t_{\mathrm{cool}}$ it would take to create this number of photons equals
(33)$$\begin{array}{ccc} t_{\mathrm{cool}} & = & \frac{N_{\mathrm{photons}}}{N_{\mathrm{atoms}}\;I}\;, \end{array}$$
where *I* denotes the average single-atom photon emission rate and $N_{\mathrm{atoms}}$ is the number of atoms involved in the cooling process. When combining the above equations, we find that the cooling rate $\gamma_{\mathrm{cool}}=t_{\mathrm{cool}}/\Delta T$ of the proposed cooling process equals
(34)$$\begin{array}{ccc} \gamma_{\mathrm{cool}} & = & \frac{c_{\mathrm{water}}\left(T\right)\;m_{\mathrm{water}}}{N_{\mathrm{atoms}}\;I\;\hbar \nu_{\max}} \end{array}$$
to a very good approximations.

As an example, suppose we want to cool one cubic micrometer of water ($V_{\mathrm{water}}=1\;\mu m^{3}$) at room temperature ($T_{0}=20$ °C). In this case, $m_{\mathrm{water}}=10^{-15}$ g and $c_{\mathrm{water}}\left(T_{0}\right)=4.18$ J/gK to a very good approximation. Suppose ν=100 MHz (a typical frequency in ion trap experiments is ν=10 MHz), $I=10^{6}/$s and $N_{\mathrm{atoms}}=10^{8}$ (a typical bubble in single bubble sonoluminescence contains about $10^{8}$ atoms). Substituting these numbers into Equation (Equation 33) yields a cooling rate of
(35)$$\begin{array}{ccc} \gamma_{\mathrm{cool}} & = & 3.81\;\mathrm{ms}/K. \end{array}$$
Achieving cooling rates of the order of Kelvin temperatures per millisecond seems therefore experimentally feasible. As one can see from Equation (Equation 33), to reduce cooling rates further, one can either reduce the volume that requires cooling, increase the number of atoms involved in the cooling process or increase the trapping frequency $\nu_{\max}$ of the atomic gas inside collapsing bubbles. All of this is, at least in principle, possible.

## 5. Conclusions

In this paper, we point out similarities between quantum optics experiments with strongly confined atomic particles and single bubble sonoluminescence experiments [29,30]. In both situations, interactions are present, which can be used to convert thermal energy very efficiently into light. When applying an external cooling laser to cavitating bubbles, as illustrated in Figure 1, we therefore expect a rapid transfer of heat into light which can eventually result in the cooling of relatively small devices. Our estimates show that it might be possible to achieve cooling rates of the order of milliseconds per Kelvin temperatures for cubic micrometers of water. The proposed quantum heat exchanger is expected to find applications in research experiments and in micro- and nanotechnology. A closely related cooling technique, namely laser cooling of individually trapped ions, already has a wide range of applications in quantum technology [9,10,11,12,13,14,15,16].

## Figures and Tables

**Figure 1 entropy-22-00379-f001:**
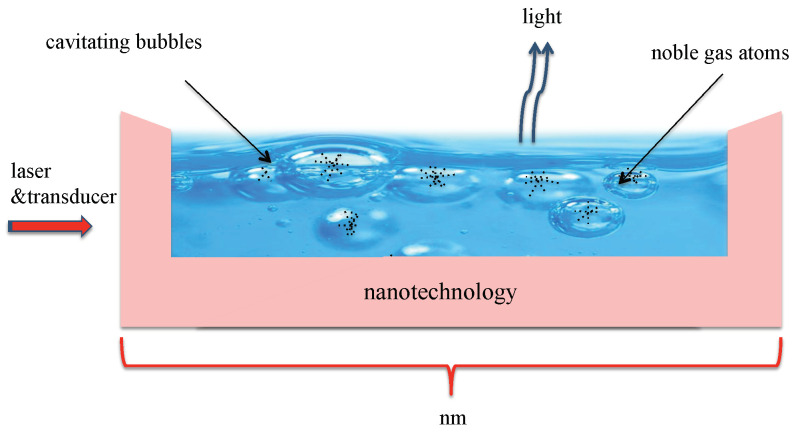
Schematic view of the proposed quantum heat exchanger. It consists of a liquid in close contact with the area which we want to cool. The liquid should contain cavitating bubbles which are filled with atomic particles, such as nitrogen, and should be driven by sounds waves and laser light. The purpose of the sound waves is to constantly change bubble sizes. The purpose of the laser is to convert thermal energy during bubble collapse phases into light.

**Figure 2 entropy-22-00379-f002:**
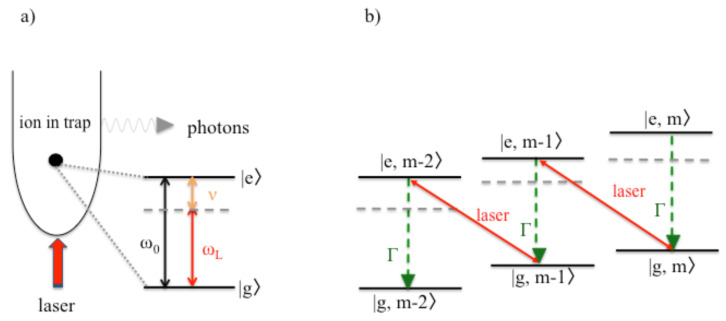
(**a**) Schematic view of the experimental setup for laser cooling of a single trapped ion. Here, |g〉 and |e〉 denote the ground and the excited state of the ion, respectively, with transition frequency $\omega_{0}$ and spontaneous decay rate Γ. The motion of the particle is strongly confined by an external harmonic trapping potential such that it quantum nature can no longer be neglected. Here, ν denotes the frequency of the corresponding phonon mode and $\omega_{L}$ is the frequency of the applied cooling laser. (**b**) The purpose of the laser is to excite the ion, while annihilating a phonon, thereby causing transitions between the basis states |x,m〉 with x=g,e and m=0,1,… of the atom–phonon system. If the excitation of the ion is followed by the spontaneous emission of a photon, a phonon is permanently lost, which implies cooling.

**Figure 3 entropy-22-00379-f003:**
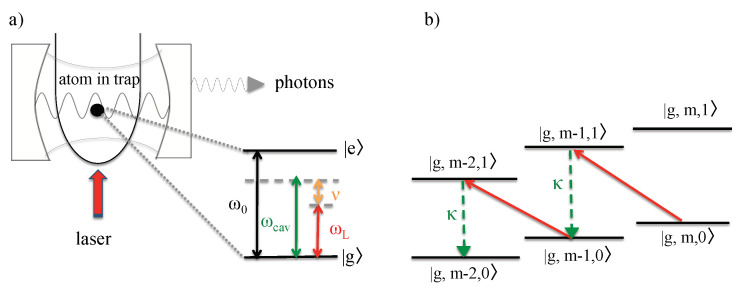
(**a**) Schematic view of the experimental setup for cavity-mediated laser cooling of a single atom. The main difference between this setup and the setup shown in Figure 2 is that the atom now couples in addition to an optical cavity with frequency $\omega_{\mathrm{cav}}$ and the spontaneous decay rate κ. Here, both the cavity field and the laser are highly detuned from the atomic transition and the direct excitation of the atom remains negligible. However, the cavity detuning $\Delta_{\mathrm{cav}}=\omega_{\mathrm{cav}}-\omega_{L}$ should equal the phonon frequency of the trapped particle. (**b**) As a result, only the annihilation of a phonon accompanied by the simultaneous creation of a cavity photon are in resonance. In cavity-mediated laser cooling, the purpose of the laser is to convert phonons into cavity photons. The subsequent loss of this photon via spontaneous emission results in the permanent loss of a phonon and therefore in the cooling of the trapped particle.

**Figure 4 entropy-22-00379-f004:**
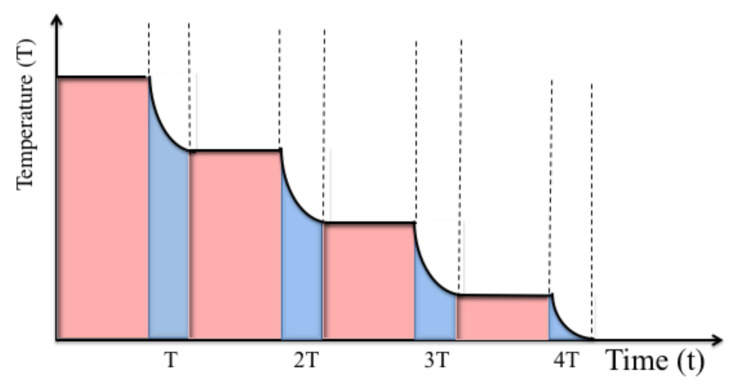
Schematic view of the expected dynamics of the temperature of the atomic gas during cavity-mediated collective laser cooling which involves a sequence of cooling stages (blue) and thermalization stages (pink). During thermalization stages, heat is transferred from the different vibrational degrees of freedoms of the atoms into a certain collective vibrational mode *B*, while the mean temperature of the atoms remains the same. During cooling stages, energy from the *B* mode into light. Eventually, the atomic gas becomes very cold.

**Figure 5 entropy-22-00379-f005:**
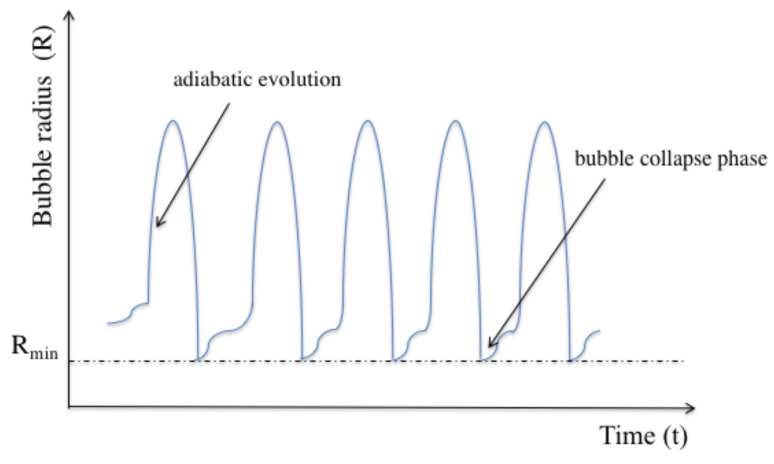
Schematic view of the time dependence of the bubble radius in a typical single-bubble sonoluminesence experiment. Most of the time, the bubble evolves adiabatically and exchanges thermal energy with its surroundings. However, at regular time intervals, the bubble radius suddenly collapses. At this point, the bubble becomes thermally isolated. When it reaches its minimum radius, the system usually emits a strong flash of light in the optical regime.

**Figure 6 entropy-22-00379-f006:**
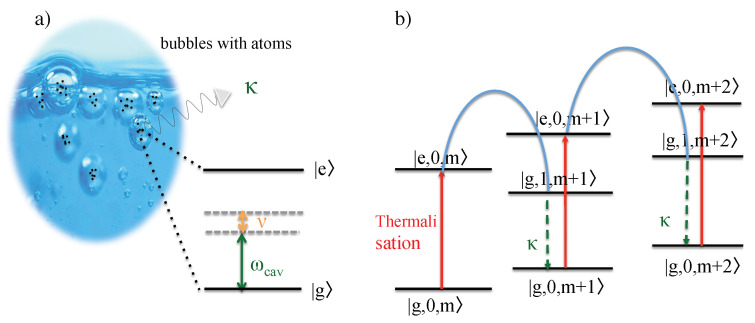
(**a**) From a quantum optics point of view, one of the main characteristics of sonoluminescence experiments is that cavitating bubbles provide a very strong confinement for atomic particles. This means that the quantum character of their motional degrees of freedom has to be taken into account. As in ion trap experiments, we denote the corresponding phonon frequency in this paper by ν. Moreover, during its collapse phase, the surface of the bubble becomes opaque and confines light, thereby forming an optical cavity with frequency $\omega_{\mathrm{cav}}$ and a spontaneous decay rate κ. (**b**) Even in the absence of external laser driving, some of the atoms are initially in their excited state |e〉 due to being prepared in a thermal equilibrium state at a finite temperature *T*. When returning into their ground state via the creation of a cavity photon, which is only possible during the bubble collapse phase, most likely a phonon is created. This creation of phonons implies heating. Indeed, sonoluminescence experiments often reach relatively high temperatures [35,36].

**Figure 7 entropy-22-00379-f007:**
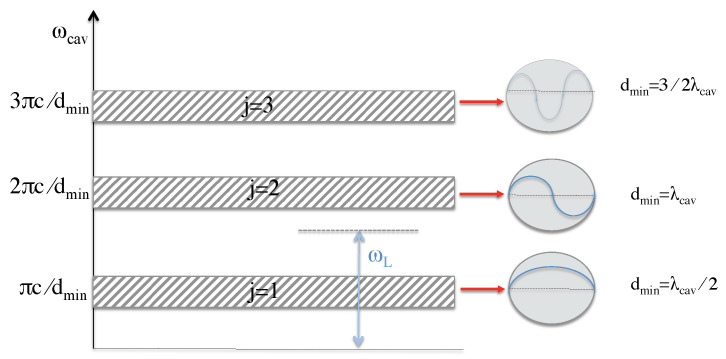
When the cavitating bubbles inside the liquid reach their minimum diameters $d_{\min}$, their walls become opaque and trap light on the inside. To a very good approximation, they form cavities which can be described by spontaneous decay rates κ and cavity frequencies $\omega_{\mathrm{cav}}$ (cf. Equation (Equation 28)). Suppose the diameters of the bubbles inside the liquid occupy a relatively small range of values. Then, every integer number *j* in Equation (Equation 28) corresponds to a relatively narrow range of cavity frequencies $\omega_{\mathrm{cav}}$. Here, we are especially interested in the parameter *j* for which the $\omega_{\mathrm{cav}}$’s lie in the optical regime. When this applies, we can apply a cooling laser with an optical frequency $\omega_{L}$ which can cool the atoms in all bubbles. Some bubbles will be cooled more efficiently than others. However, as long as the relevant frequency bands are relatively narrow, none of the bubbles will be heated.

**Figure 8 entropy-22-00379-f008:**
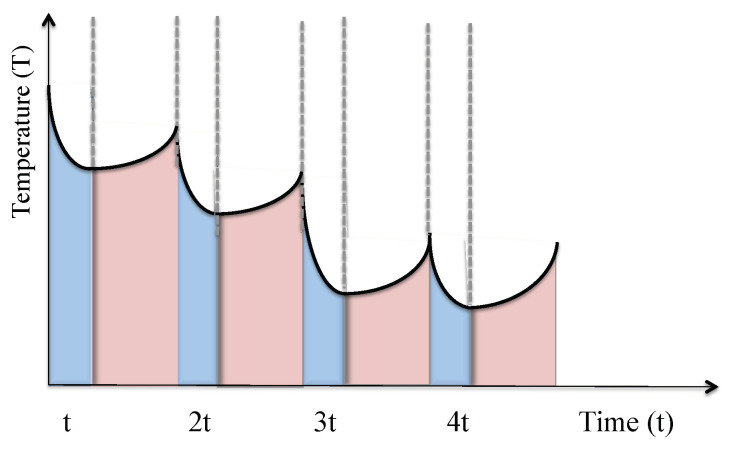
Schematic view of the expected dynamics of the temperature of a confined atomic gas during bubble collapse stages (blue) and expansion stages (pink). During expansion stages, heat is transferred from the outside into the inside of the bubble, thereby increasing the temperature of the atoms. During bubble collapse stages, heat is converted into light, thereby resulting in the cooling of the system in Figure 1. Eventually, both processes balance each other out and the temperature of the system remains constant on a coarse grained time scale.

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
