# Peer review of "A Quantum Heat Exchanger for Nanotechnology"

_entropy, 2020, doi:10.3390/e22040379_

Round 1

Reviewer 1 Report

This is an interesting and curious paper. The authors propose a quantum heat exchanger which converts heat into light using cavity-enhanced laser cooling of an atomic gas inside a cavitating bubble. Heat exchangers in nanotechnology usually rely on fluid flow. Here the authors propose instead to use a variation of the well established quantum-optical technique of laser cooling, namely cavity-mediated collective laser cooling. The quantum heat exchanger is a liquid which contains a large number of cavitating bubbles filled with noble gas atoms. Transducers constantly change the radius of these bubbles to make them resemble optical cavities when they reach their minimum radius during bubble collapse. At this point, an appropriately detuned external laser field is applied in addition to the sound waves which confine the bubbles. The laser field transfers vibrational energy of the atoms into light. If the surrounding liquid contains many cavitating bubbles, their surface area becomes relatively large and there can be an efficient exchange of heat between the inside and the outside of the cavitating bubbles.

The authors assert that this proposal is experimental feasible. While I am a little skeptical in this claim I believe that this paper will spark interest in the community. For this reason I am happy to recommend publication.

Author Response

We would like to thank Referee 1 for their supportive comments. In response to the referee comments, we clarified some of our explanations and added some references to sonoluminescence and to quantum optics papers which support our hypothesis that both areas of physics are closely connected. 

Reviewer 2 Report

The authors discuss possibilities of sympathetically cool a gas bubble in a liquid via laser cooling of atoms' motion. I am not aware of exactly this type of proposal so in that sense there is an aspect of novelty. However, there are issues.

I think the authors have to make things quantitative given some reasonable experimental parameters. Otherwise reader cannot tell if the idea makes any sense.
- What is the frequency distribution of atom vibrations inside the collapsing bubble? If there are barely no atoms near the red-sideband frequency, no cooling is produced. Given the mean free path in a gas at ambient is about 100 nm, I would expect the atoms not care about any trapping potentials beyond this length scale.
- How much is the temperature of the gas bubble reduced during the cooling?
- How much heat is generated during the sonoluminescence event? Compared to the cooling power you expect?
- give values for g_eff and cooling rate.
- compute needed laser power.

In some sense I appreciate the pedagogical approach, but most of it is empty textbook material. Therefore the paper is far too long given the (missing) depth of the discussion. It could (and probably should) be shortened down to say 25% in order to fit a quality standard for a common scientific paper.
- At least remove discussion of partition function of an oscillator, just state the BE thermal photon number in one sentence.
- remove Lindblad formulas.

Smaller comments:
- "becomes quantized". All oscillators are quantized, it is rather when the quantization is observable. I don't think it anyway makes sense to speak of quantization of the motion when the modes are somewhere around room temperature - the phonon numbers are probably (missing number) quite astronomical.
- "so-called phonon". Do you expect to have readers who have never heard of a phonon?
- "collective enhancement" is quite misleading since the relative change of temperature of an ensemble (one atom or much more) is presumably not affected by having many atoms.

In summary, without addressing the above points my opinion is that the paper is empty and cannot be published.

Author Response

We would like to thank the referee for acknowledging the novelty of our work. However, the referee also has some issues with our work. Previously, there have been only very few attempts to look at cavitating bubbles from a quantum optics point of view. Therefore very little is known about experimental parameters, like the distribution of atom vibrations inside collapsing bubbles. However, many of the observations made in standard sonoluminescence experiments provide a strong indications that quantum optics effects play a crucial role. During each bubble collapse phase, a strong light flash occurs which supports our hypothesis of a sufficiently strong confinement of an atomic gas.

Also, it is important to notice that laser cooling does not require phonon frequencies to coincide exactly with laser detunings. Resonance is only a condition for maximum cooling. Cavity-mediated laser cooling occurs as long as the frequency of the applied laser field is to below the cavity frequency. In this case, the missing energy is taken from the motion of the atoms.

Unfortunately, we cannot make more accurate predictions concerning the temperature change of the atomic gas inside cavitating bubbles. However, as pointed out in Section 4, we expect that every photon emitted during the bubble collapse phase removes one phonon from the system. If the number of emitted photons is sufficiently high, then there can be a significant cooling effect. The fact that laser cooling can transfer single ions to nanoKelvin temperatures suggests that the proposed heat exchanger can result in significant cooling rates also in slightly bigger setups. 

The main difference between sonoluminescence experiments and cavitating bubbles is laser driving. In sonoluminescence experiments, the interaction between phonons and photons is highly detuned. The dynamics of the system is therefore dominated by the simultaneous creation of cavity photons and phonons. As soon as a laser field is applied, the conversion of phonons into cavity photons becomes resonant or almost resonant. We therefore expect that the conversion of phonons into cavity photons dominates the dynamics of the system, while processes which occur during sonoluminescence experiments and result in heating become negligible.  

The purpose of our paper is to point out that the questions which have been raised by the referee are not only of theoretical but also of practical interest and to inspire quantum optics experiments with cavitating bubbles. A crude estimate which borrows data from different, already available experiments suggests that it might be possible to achieve cooling rates of the order of Kelvin temperatures per millisecond for volumes of liquid on a cubic micrometer scale. 

Concerning the additional comments of Referee 2 (second paragraph of their report):

* We would like to thank the referee for acknowledge our pedagogical approach. The paper does not aim at researchers, who are already familiar with laser cooling, but at scientists working with cavitating bubbles and in nanotechnology. As suggested by Referee 2, we shortened the text of the paper by about 1.5 pages. Shortening the paper more seemed counter productive. A pointed out by Referee 1, “To my opinion, the presentation of the work is clear. The authors make an effort in introducing cavity-mediated laser cooling in a pedagogical way, which I find it very useful for the non-expert reader.“

* The discussion of the partition function of a single harmonic oscillator has been shortened significantly, as proposed by the referee.

* The Lindblad formulae is important, since our approach to cavity-mediated collective laser cooling is new. Previously, it has not been shown that this form of laser cooling only affects a single collective vibrational mode. The equations which we show here have been carefully chosen and are needed to provide a justification for our statements.

* The formulations “their motion become quantised” and  “so-called phonons” have been replaced everywhere.

* The term “collective enhancement” in our paper refers to the fact that the effective cooling rate of N atoms is N times the size of the effective cooling rate of a single atom. This fact is now more clearly explained in the text.  

In response to the comments of the referee, we carefully rewrote some explanations. In addition, we added some references to papers which discuss closely related  issues and support the feasibility of the cooling scheme which we discuss here.

Reviewer 3 Report

In this manuscript, the authors propose a quantum heat exchanger for applications in micro and nanotechnology. The heat exchanger uses cavity-mediated laser cooling of an atomic gas trapped in cavitating bubbles, which allows for conversion of heat into light. By employing ultrasonic transducers in a liquid contacting the surface of the device to be cooled down, it is possible to generate and control the size of the cavitating bubbles. When the bubbles reach their minimum size, these act similar to optical cavities confining the atomic gas in such a way their motion becomes quantised, thus enabling resonant conversion of vibrational energy into light through an applied laser. Through the reduction of the phonon population in a specific vibrational mode during bubble collapse, together with the heat exchange of the atomic gas with its
surrounds during adiabatic expansion, it is possible to cool down the surrounding liquid and, with it, the contacted device’s surface.

To my opinion, the presentation of the work is clear. The authors make an effort in introducing cavity-mediated laser cooling in a pedagogical way, which I find it very useful for the non-expert reader. The proposed heat exchanger is then explained in detail, bringing in combination quantum optics and sonoluminescence concepts. Although it is hard to tell whether the cooling rates needed for applications in micro and nanodevices would be reached by the proposed heat converter, the idea of cavitating bubbles for vibrational energy extraction is surely attractive, and would probably stimulate further experimental research.

With this said, I recommend this manuscript for publication in Entropy, though I have a few minor comments that the authors may consider in a revised version of the manuscript:

  • Line 59. Perhaps at this point the authors should briefly explain what is meant by sympathetic cooling.

  • Line 73. The phrase:

    “In the following, we cannot provide accurate estimates, not even orders of magnitude, for the relevant system parameters of the experimental setup shown in Fig. 1. Nevertheless, we estimate that it seems feasible to achieve cooling rates of the order of Kelvin temperatures per millisecond for volumes of liquid on a cubic micrometer scale.”

    sounds to me a bit contradictory, since in the end of the manuscript the authors estimate (very roughly) some possible values for the cooling rates. Perhaps they should clarify why in general it is not possible to provide such estimates.

  • Line 115. The phrase:

    “If the excitation of the atom is followed by the spontaneous emission of a photon, the atom returns into its ground state without changing its phonon number.”

    was not clear to me in this paragraph where the authors are explaining the cooling process.

  • Line 121. Shouldn’t be “... accompanied by the annihilation of a phonon” or “... accompanied by the creation of a photon” instead?

  • In Eq. (11), I believe that the rho’s appearing within the parenthesis should be replaced by rho_I.

  • Perhaps it is more instructive to write Eq. (19) exactly in the same way as Eq. (10), i.e.,

    H_I = \hbar \tilde{g}_\mathrm{eff} \left( B^\dag c + B c^\dag \right)
  • Lines 278 and 279. The authors should improve a little bit these two sentences, as this was already introduced in the previous sections. They should use something like: “As we already pointed out, the purpose of this paper is ...”

Author Response

We would like to thank Referee 3 for enthusiastically recommending the publication of our manuscript. Their comments have been very thoughtful and helped us to improve our manuscript significantly. All of the suggestions of the referee have been taken into account.

* Line 59: The term “sympathetic cooling” has been removed from the paper. Instead we explain that the bubble evolves adiabatically during its expansion phase and heat is transferred from the surroundings of the bubble to the atomic gas inside.

* Line 73. The phrase has been replaced. We now write, “Although sonoluminescence has been studied in great detail and the idea of applying laser fields to cavitating bubbles is not new [34], not enough is known about the relevant quantum properties, like phonon frequencies. Hence we cannot predict realistic cooling rates for the experimental setup shown in Fig. 1. However, a crude estimate which borrows data from different, already available experiments suggests that it might be possible to achieve cooling rates of the order of Kelvin temperatures per millisecond for volumes of liquid on a cubic micrometer scale.” 

* Line 115: The statement has been rewritten and the explanation of the cooling process has been improved. 

* Line 121: This mistake has been corrected.

* Eq. (11): This typo has been corrected. 

* Eq. (19): We followed the recommendation of the referee and re-wrote the equation. 

* Line 278 and 279: The two sentences have been shortened.

Round 2

Reviewer 3 Report

First of all, I agree with Referee 1 in that this work somehow lacks of a concrete model able to provide good estimates for all relevant quantities, like cooling rates, etc. However, as I reported before, the idea of laser cooling with cavitating bubbles seems to me be very attractive, and surely would stimulate further experimental research. Along this direction, Ref. [36] of the revised version points towards this topic (use of laser in sonoluminescing bubbles), but the size of the bubbles is not enough as to ensure a discrete phonon spectrum. I wonder then whether quantum confinement would be possible to observe in these samples, since this seems to be one of the loose points in this proposal. Any effort in clarifying this will indeed strengthen the manuscript.

On the other hand, the changes made by the authors surely improve the overall reading of the manuscript. Therefore, I believe that the revised version, together with a more detailed discussion about quantum effects in cavitating bubbles, is well suited for publication in Entropy.

Author Response

We would like to thank Referee 3 for having another look at our manuscript and for recommending the publication of our work. We agree with the referee that applying laser pulses to cavitating bubble might simply not affect their temperature (cf. e.g. Ref. [36]). As we explain in our manuscript, we only expect laser cooling when the frequency of the applied laser field does not exceed the cavity frequency associated with the minimum bubble radius. Laser parameters need to be chosen carefully. In addition, the relevant phonon frequency of the confined atomic species needs to be sufficiently large (cf. Eq. (2)). It would therefore be helpful to know more about this parameter.

In reply to the comment of the referee, we changed the last paragraph on page 2 and the first paragraph on page 3 of the Introduction. The paper now states more clearly that the pressure at the location of the bubble can be significantly larger than the applied external driving pressure [32]. However the strongest indication for the presence of phonon modes with sufficiently large frequencies for laser cooling to work comes from the fact that sonoluminescence experiments are well-known for converting sound into relatively large amounts of thermal energy, while producing light in the optical regime [33–35]. During this process, the atomic gas inside a cavitating bubbles can reach very high temperatures [36,37] which hints at very strong couplings between electronic and vibrational degrees of freedom.